# Technical note: Space-time analysis of rainfall extremes in Italy: clues from a reconciled dataset

Andrea Libertino[1], Daniele Ganora[1], and Pierluigi Claps[1]

[1]Department of Environment, Land and Infrastructure Engineering (DIATI), Politecnico di Torino, Torino, ITALY.

**Correspondence:** Andrea Libertino (andrea.libertino@polito.it)

**Abstract.** Like other Mediterranean areas, Italy is prone to the development of events with significant rainfall intensity, lasting for several hours. The main triggering mechanisms of these events are quite well known but the aim of developing rainstorm hazard maps compatible with their actual probability of occurrence is still far from being reached. A systematic frequency analysis of these occasional highly intense events would require a complete countrywide dataset of sub-daily rainfall records, but this kind of information was still lacking for the Italian territory. In this work several sources of data are gathered, for assembling the first comprehensive and updated dataset of extreme rainfall of short duration in Italy. The resulting dataset, referred to as Italian Rainfall Extreme Dataset (*I-RED*), includes the annual maximum rainfalls recorded in 1 to 24 consecutive hours from more than 4500 stations across the country, spanning the period between 1916 and 2014. A detailed description of the spatial and temporal coverage of the *I-RED* is presented, together with an exploratory statistical analysis aimed at providing preliminary information on the climatology of extreme rainfall at the national scale. Due to some legal restrictions, the database can be provided only under certain conditions. Taking into account the potentialities emerging from the analysis, a description of the ongoing and planned future work activities on the database is provided.

## 1 Introduction

Italy can boast of a role at the highest level in the development of meteorological observations (Brunetti et al., 2006), with 6 meteorological stations operating since the eighteenth century (Bologna, Milano, Roma, Padova, Palermo and Torino), and 15 stations with observation starting in the first half of the nineteenth century. First attempts of performing a systematic collection of monthly rainfall data go back to as early as 1880 when the National Office for Meteorology and Climate was funded. The National Hydrographic Service (*SIN*) and the National Hydrographic and Mareographic Service (*SIMN*) collected annual maxima values for 1-3-6-12 and 24 hours durations in the Hydrological Yearbooks from 1917 to early 2000s (the final publication year depends on the local agencies of the *SIMN*). The D.Lgs 112/1998 dismantled the *SIMN*, transferring its tasks to the 19 administrative regions and the 2 autonomous provinces of Trento and Bolzano. These authorities were designated as local Operational Centres and Regional Environmental Agencies to deal with hydro-meteorological monitoring and civil protection issues.

Inspite of the huge heritage of data, only a small fraction of the Italian rainfall data is available in a computer-readable format. Moreover, the dismantlement of the National Service led to a lack of update of the national database of extreme

rainfall that is still stuck, for some regions, at the beginning of the '90s. This has led to a very fragmented framework: updated rainstorm hazard assessments are actually only available for some regions and only at the regional scale (see, e.g., Uboldi et al. (2014); Libertino et al. (2017)). Various regional studies present different methodologies and are sometimes based on very different data densities and record lengths (e.g., Claps et al., 2016), but only few updated analyses on short-duration rainfall in 5 a over-regional framework are available (e.g., Rudari et al., 2005; Allamano et al., 2009).

In view of the assembling of the first comprehensive dataset of extreme rainfall of short duration in Italy several major sources of data have been analysed. The resulting dataset, referred to as Italian Rainfall Extremes Database (*I-RED*), includes data from more than 4500 stations across the country, spanning the period between 1916 and 2014, and refers to annual maximum rainfall recorded in 1 to 24 consecutive hours (exact durations available are 1-3-6-12 and 24 hours).

10 The following sections describes the sources of the data, the work carried out for the merging of the database and the operations that are still required for making it suitable for a nationwide robust rainfall frequency analyses. A preliminary analysis of the extreme rainfall regime at the national scale is also presented.

## 2 Merging the *I-RED* Dataset

### 2.1 Data sources

15 As a follow-up of the activities of the Italian National Group for the Prevention of the Hydrogeological Disasters (*GNDCI*) a comprehensive nationwide hydrological information system has been set up, within the "*CUBIST* project", funded by the Italian Ministry of Education and Research within the call *PRIN* 2005 (Italian Research Projects of National Relevance). The database includes about 6000 pluviographs and pluviometers, 700 temperature stations and about 400 river basins (Claps et al., 2008) and is available at: http://www.cubist.polito.it; accessed: 2017-10-26. In the detail, the database includes rainfall data 20 from 1900 to 2001 (depending on the region) and constitute the first important attempt of making the large Italian hydrological heritage freely available in a computer-readable format. The annual maxima data for different durations included in the *CUBIST* dataset are extracted with a sliding-windows process from manual tipping-bucket rain gauge data, equipped with a recording system that writes on diagram paper. Further information on the characteristics of the stations can be found in Acquaotta et al. (2016). The number of data per year is not constant across the analysed period, being increasing in time as more stations have 25 been installed in the recent years. Data availability decreases in the period of the Second World War, as many records have been missed in that period. After 1980, with the progressive dismissal of the *SIMN* and the development of the local hydrographic authorities, data availability decreases rapidly until 2001, when the rain gauges still under the *SIMN* were taken over by the local Operational Centres.

After the late '80s, indeed, the local Environmental Agencies started to support the *SIMN* in its work. Gradually, the 21 30 regional hydrological services took over the networks and the tasks of the national one. In this period most of the old manual tipping-bucket rain gauges have been substituted with automatic stations, similar to the one described in Acquaotta et al. (2016) for the Piemonte region. Each hydrological service adopted its own rules for the publication of the collected data and, even if the Italian law adopted for the public data an open source policy for non-commercial uses (under D.Lgs.82/2005, D.Lgs.36/2006,

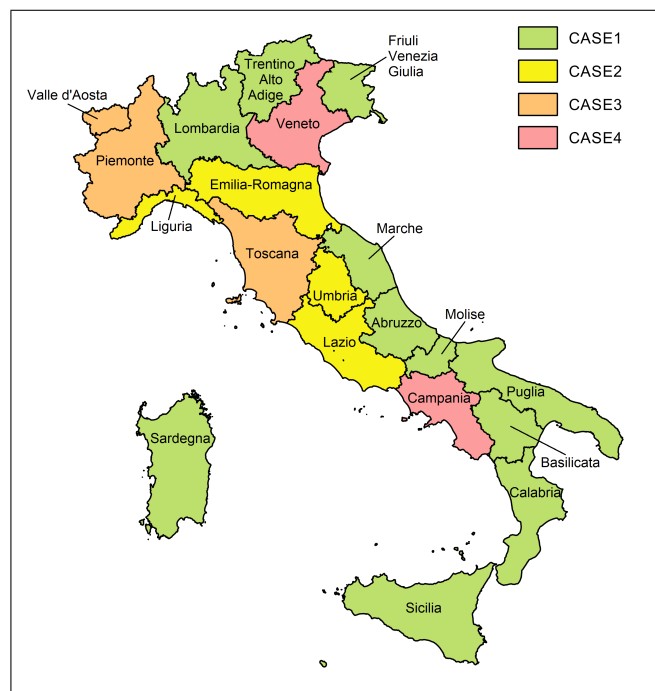

**Figure 1.** Names of the Italian regions and type of datasets provided by the regional authorities. The cases refer to the bullet list of section 2.2.

D.M.10/11/2011, L.221/2012, D.Lgs.179/2012 and L.114/2014), an updated database of the annual rainfall maxima for sub-daily duration at the national scale is still lacking. For the scopes of this research, the different agencies have been contacted, requesting the regional annual maxima datasets for sub-daily durations. The regions of Italy are shown in Figure 1 together with the type of data provided, that will be described in the following section. Table 1 lists the name of the local authorities and the regional codes, aimed at identifying them in the database. The public availability of the original dataset is also reported.

## 2.2 Cleaning and merging operations

Merging and harmonizing the different datasets is a quite long and difficult operation, that is still ongoing. The different Operational Centres provided different types of datasets, with different temporal coverages and spatial reference systems. Duplicate stations are often present in the databases of neighbouring regions.

The first steps of this work have been carried out at the regional scale. For each region all the data falling inside the regional boundaries have been considered. These data, according to the setting of the databases of the local Operational Centers, could belong to one of these 3 categories:

a) Data from the *CUBIST* database for the 1900-2001 period already available from the former national service

b) Data provided by the regional authority

**Table 1.** Regions of Italy with the assigned code and the related local Operational Center with references to the availability of digitized data.

| CD | Region | Operational Center | Digitized data availability |
|---|---|---|---|
| 01 | Abruzzo | Ufficio Idrografico e Mareografico Regione Abruzzo | Available upon request |
| 02 | Basilicata | Dipartimento Protezione Civile Regione Basilicata | Available in [1] |
| 03 | Calabria | Centro Funzionale Multirischi - ARPACAL | Available at [2] |
| 04 | Campania | Centro Funzionale Regione Campania | Available upon request |
| 05 | Emilia-Romagna | ARPA Emilia-Romagna | Available upon request |
| 06 | Friuli Venezia Giulia | Ufficio Idrografico Regione Autonoma Friuli Venezia Giulia | Available upon request |
| 07 | Lazio | Centro Funzionale Regione Lazio | Available upon request |
| 08 | Liguria | ARPAL-CFMI-PC | Partially available at [3] |
| 09 | Lombardia | ARPA Lombardia | Available at [4] |
| 10 | Marche | Dipartimento di Protezione Civile Regione Marche | Available at [5] |
| 11 | Molise | Centro Funzionale Regione Molise | Available upon request |
| 12 | Piemonte | ARPA Piemonte | Partially available at [6] |
| 13 | Puglia | Dipartimento di Protezione Civile Regione Puglia | Available at [7] |
| 14 | Sardegna | ARPAS | Available upon request |
| 15 | Sicilia | Osservatorio delle Acque Regione Siciliana | Available upon request |
| 16 | Toscana | Servizio Irdrografico Regionale Toscana | Available at [8] |
| 17 | Trento * | Centro Funzionale Provincia Autonoma di Trento | Available at [9] |
| 18 | Bolzano - Alto Adige * | Ufficio Idrografico Provincia Autonoma di Bolzano - Alto Adige | Available upon request |
| 19 | Umbria | Regione Umbria | Available upon request |
| 20 | Valle d'Aosta | Centro Funzionale Regione Autonoma Valle d'Aosta | Available upon request |
| 21 | Veneto | ARPAV | Available upon request |

* the Autonomous Provinces of Trento and Bolzano - Alto Adige, together, constitute the region Trentino Alto Adige (CD: 22)

[1] Manfreda, S., Sole, A. and De Costanzo, G.: Le precipitazioni estreme in Basilicata, Editrice Universo Sud, 2015.

[2] ARPACAL: Centro Funzionale Multirischi, http://www.cfd.calabria.it/, accessed: 2016-08-01

[3] ARPAL: Consultazione Dati Meteoclimatici, http://www.cartografiarl.regione.liguria.it/SiraQualMeteo/Fruizione.asp, accessed: 2016-08-01

[4] ARPA Lombardia: Progetto Strada, http://idro.arpalombardia.it/pmapper-4.0/map.phtml, accessed: 2016-08-01

[5] Protezione Civile Regione Marche: Annali Idrologici Regione Marche, http://console.protezionecivile.marche.it, accessed: 2016-08-01

[6] ARPA Piemonte: Banca dati meteorologica, http://www.regione.piemonte.it/ambiente/aria/rilev/ariaday/annali/meteorologici, accessed: 2016-08-01

[7] Protezione Civile Puglia: Annali Idrologici - Parte I, http://www.protezionecivile.puglia.it/centro-funzionale/analisielaborazione-dati, accessed: 2016-08-01

[8] SIR Toscana: Settore Idrologico Regionale, http://www.sir.toscana.it/, accessed: 2016-08-01

[9] Centro Funzionale di Protezione Civile Provincia Autonoma di Trento: Meteotrentino, http://www.meteotrentino.it/, accessed: 2016-08-01

c) Data provided by the regional authorities of the neighbouring regions, falling out of their regional borders

Observations dating before 1916 have been discarded, as considered not significant and too unevenly distributed. Considering that most of the provided data have been validated from the related authorities, they are considered reliable and, at first, included directly in the *I-RED*. For information on the validation procedures, please refer to the Appendix A and to Barbero S. et al.

(2017). In the presence of inconsistencies between the type b) and type c) data, preliminary manual merging was carried out. The sources of the inconsistencies could be various, according to the evolution of the monitoring systems of the different regions and often is due to the joint management of interregional basins. The different regional authorities often have adopted different codes/names for the same station, the first step has been thus to identify the presence of duplicate stations with same/similar name covering different time intervals. Sometimes, even for the same station, neighbouring regions can provide different data for the same years. This can be, e.g., due to the fact that sometimes regions share rainfall data before their validation and official publication. If the same station was found in the database of more neighbouring regions a first attempt of merging the series together was carried out, by analysing the data recorded year by year. If the merging was not feasible, higher priority was given to the data provided by the authority of the considered region (that is usually also the owner of the network). This allowed to avoid the presence of duplicate series in the *I-RED*.

Once merged, for each region, type b) and type c) datasets, the resulting dataset has to be merged with the type a) dataset. This operation has been quite complex, as the overlapping period between the different dataset was different for each region and because most of the authorities did not tracked the change in the name/code of the stations. The different procedures performed, according to the type of the dataset that the region has provided (as reported in Figure 1) can be summarized as follows:

1. *Regions that digitized the whole SIMN database for their areal domain and provide a complete merged database.* The provided data were inserted in the *I-RED* without editing and without considering the *CUBIST* series. Only for the Abruzzo and Molise regions some preliminary refinement was needed, as the two regions were divided in 1963, and the databases of the two regions partially overlap. The stations were then divided according to the actual regional boundaries and the duplicate series removed.

2. *Regions that provided datasets including data from their actual regional network partially merged with subsets of digitized data from the SIMN Hydrological Yearbooks.* As not all the *SIMN* datasets were digitized from the local authorities, the dataset lacked part of the stations included in the *CUBIST* database. To maximize the available information, data from the regional databases and the *CUBIST* one were manually analyzed and merged, in order to avoid duplicate values. Stations with same name and similar coordinates were merged together in the presence of a two-year consistent overlapping period. In the presence of inconsistencies between the values recorded by the two stations two stations in the overlapping period, they were treated as different stations and renamed. If that was not possible to unravel any doubt, the stations were considered as separate entities. For the Liguria region, the information in ARPAL (2013) was used to overcome the lack of information on the continuity of the series.

3. *Regions that provided two different datasets: one containing the whole digitized data from the SIMN stations and another containing the digitized data from their actual networks.* The data of the two databases were merged together, the overlapping period manually analysed to avoid overlapping, and the *CUBIST* database ignored. The operation was made possible by the collaboration of $ARPA$ Piemonte, for Piemonte and Valle d'Aosta, and of the Università degli Studi di Firenze, for Toscana.

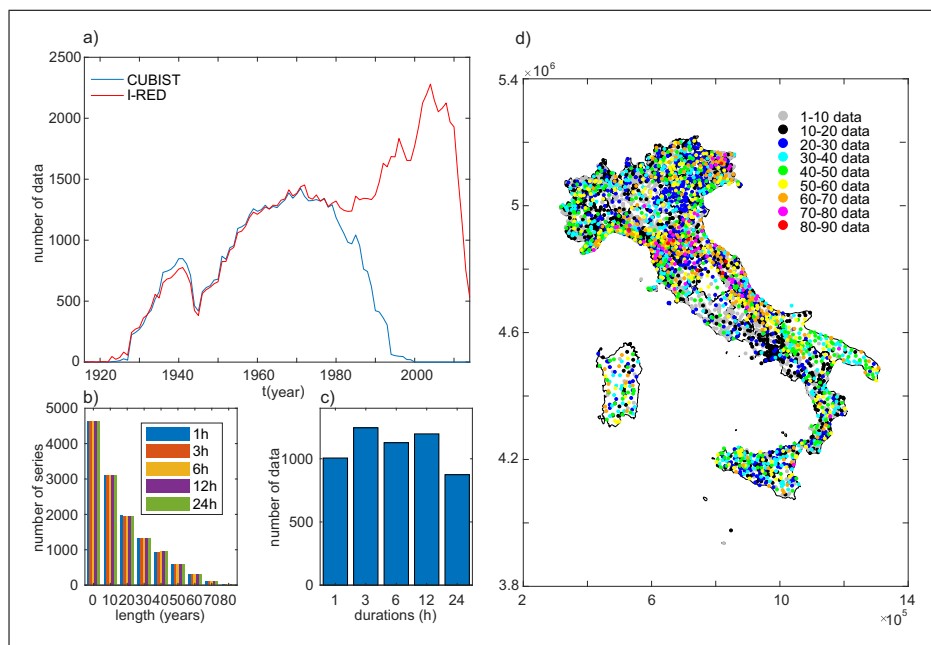

**Figure 2.** (a) Data availability per year in the *I-RED* and *CUBIST* databases (the smallest value across the 5 considered duration is reported per each year). (b) Number of series longer than fixed threshold values in the *I-RED* databases per duration (null values are ignored). (c) Number of null values per duration. (d) Length of the series in the *I-RED* database represented in space: the color refers to the minimum length among the 5 available durations. If more stations overlap due to the resolution of the picture, the one with the longer series appears on the top.

4. *Regions that provided only the data recorded from the network they actually manage.* All the information concerning the *SIMN* stations was lacking. The provided dataset was therefore merged with the whole *CUBIST* database for the considered regions. Duplicate values were excluded analysing manually the overlapping period, if present.

With the application of the above described rules, 20 complete regional datasets have been obtained. The regional datasets were finally merged together to generate the *I-RED*. After the merging phase some reliability check has been performed, in order to detect any problematic or incorrect information. They includes the identification and removal of the duplicate data/stations and reliability checks on the larger values of the dataset, comparing them to the absolute record-breaking events for all the durations (see Libertino (2017)), aimed at detecting inconsistencies in rainfall series. If any suspect value was found, its year of occurrence was compared, when referring to recent years, with the data from event reports or newspapers. If the data refers to a *SIMN* station, the Hydrological Yearbooks were consulted. If no evidence was found, the related authority was contacted. Most of the operations need human supervision, and a thorough verification work. If it is not possible to unravel any doubt the suspect value is discarded.

Due to the complexity of the check operations, further efforts and collaborations with the regional authorities are still ongoing to increase the consistency of the database. Nevertheless, to date (October 2017) the *I-RED* includes more than 4500 stations nationwide and constitute the largest updated dataset of annual maxima for Italy.

Considering that most of the regional authorities supervise the use and widespread of their datasets for preventing improper uses, a detailed description on how to access the *I-RED* is reported in the "Data availability" box.

In the following, the spatio-temporal distribution of the assembled data will be described.

## 3   Main features of the *I-RED* database

The number of data available per year in the *I-RED* is reported in Figure 2a, as compared with that of the *CUBIST* database. As every station is related to a unique value of annual maxima for a given duration, the presence of a measurement implies the presence of a station. The number of available stations increases with time, and drastically grows after the dismissal of the *SIMN* and the development of the local agencies. The decrease after 2010 can be attributed to the fact that not all the regions have published the data for the most recent years.

The smaller size of the *I-RED* compared the *CUBIST* database in some years can be due to:

– The presence, before 1945, in the *CUBIST* database of data from territories lost by Italy after World War II (e.g. , Istria) or from neighbouring countries, not included in the *I-RED*;

– The fact that regional agencies could have decided for different reasons not to include data or stations from the *SIMN* dataset in their database. Part of these data could therefore be lost not considering the *CUBIST* database for these regions.

Considering the limited significance of the information loss, further efforts for including these data will be planned only in a future stage of the development of the database.

For a descriptive analysis of the rainfall data, all the assembled time series are classified according to their length. Results are shown in Figure 2b. Considering the short life of the rain gauges installed by the regional Operational Centers, a large percentage of the series is shorter than 20 years but the contribution of the *CUBIST* database allows for a significant amount of longer series. The series with more than 80 years of data are for the 1, 3, 6, 12, 24 hours durations are respectively 16,14,15,14 and 17. In general, all the durations report a similar behaviour, despite some differences in the distribution of the null values as shown in Figure 2c. The reasons that lead to missing data only for certain durations can be various and related to either the measuring, the recording or the storage of the data (e.g., missed reading of the record from the operator, data classified as not-valid in the validation phase, etc.).

The spatial distribution of the stations is shown in Figure 2d. The color scale refers to the number of the available data per each series. The minimum number across the 5 duration is considered. One can clearly distinguish that, even if the whole national territory is represented, the density of the stations widely changes across the nation. To show the relevance of the non-uniformity, a gridded domain with a mesh size of 50 km is introduced. Figure 3 shows the number of station-year, i.e., the total number of data per cell, showing all the available data of the stations located within the cell. If data consistency changes

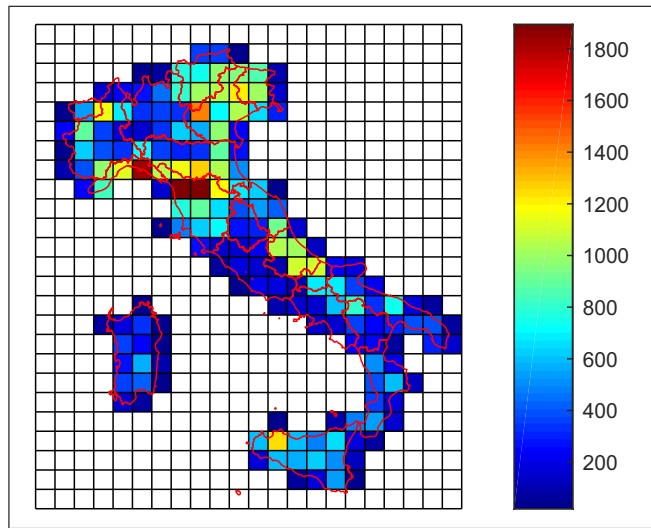

**Figure 3.** Number of station-year per cell over a 50 km grid.

for the different durations, the shortest one is considered. The non uniformity of the network density clearly emerges at first sight, with some cells presenting almost 10 times the number of data of other cells. The most densely gauged cells can be found in the North-West of the country, in particular in Liguria region, in the northern Toscana and, in the North-East.

## 4 Descriptive statistical analysis of rainstorms in Italy

A preliminary descriptive analysis of the characteristics of extreme rainfalls at the national scale has been carried out on the newly developed *I-RED* database. Series with a minimum length of 20 years of data have been considered in this analysis. This length constraint leads to a subset of 1974 series available for the analysis, out of the original 4686. For each duration, the median of the series is depicted in Figure 4. The median is used as a robust estimator of the central tendency of a series, less sensitive than the mean to the presence of outliers. As common methods of fitting distributions, e.g., product moments

or L-moments are using mean values for representing the central tendency, maps of the mean for the different durations are attached in the supplementary material.

Some geographical areas are characterized by clusters of large median values and these clusters appear consistent across the different durations. Furthermore, at the country-wide scale we observe that the coefficient of variation of the medians increases for increasing durations, suggesting a wider range of variability of the corresponding median values.

For each series, the sample L-moments (Hosking and Wallis, 1997) have then been computed to describe the shape of the empirical distribution of the records. The mean L-moments ratios among the different durations give information respectively on the dispersion (L-CV), skewness (L-skewness) and "peakedness" (L-kurtosis) of the empirical distributions. All the above statistics are mapped in Figure 4. Considering that the L-moments ratios show similar behaviour for the considered durations, we decided for simplicity to include in the paper only the average ones. The maps for the different durations are reported in the

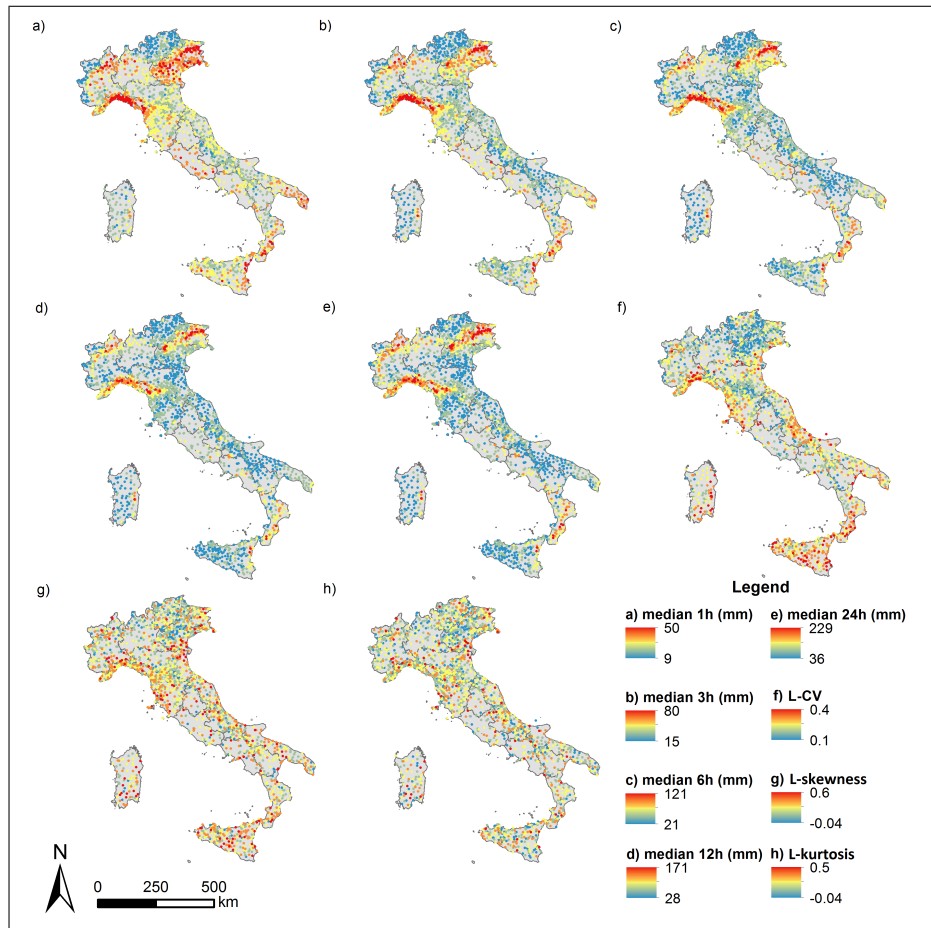

**Figure 4.** Median values of the *I-RED* series from 1 (a) to 24 (e) hours. Average statistics for the five durations considered: (f) L-CV, (g) L-skewness and (h) L-kurtosis. Series with more than 20 data are considered.

supplementary material. Figure 4f shows that the coastal areas and the islands are generally characterized by a higher variability in the annual maxima series, presenting larger L-CV values. The northern part of the peninsula, even if characterized by large median values, shows lower L-CV, which is typical of areas with large average rainfall values. It is harder to identify a precise spatial pattern in the distribution of the skewness and kurtosis values (Figures 4g and 4h). Coastal and island areas seem to generally show larger skewness values, confirming the influence of the Mediterranean sea on the climate of these areas. All the aforementioned maps have been also interpolated for visualization purposes with ordinary kriging; detailed results are reported in the supplementary material.

The significance of the developed dataset allows also to preliminary explore the rainfall events sometimes referred to as "black swans" (Blöschl et al., 2015), showing extraordinary intensities even when compared with the population of annual maxima. In Italy, many of these events have been studied as individual extraordinary events (e.g., Rebora et al., 2013; Fiori

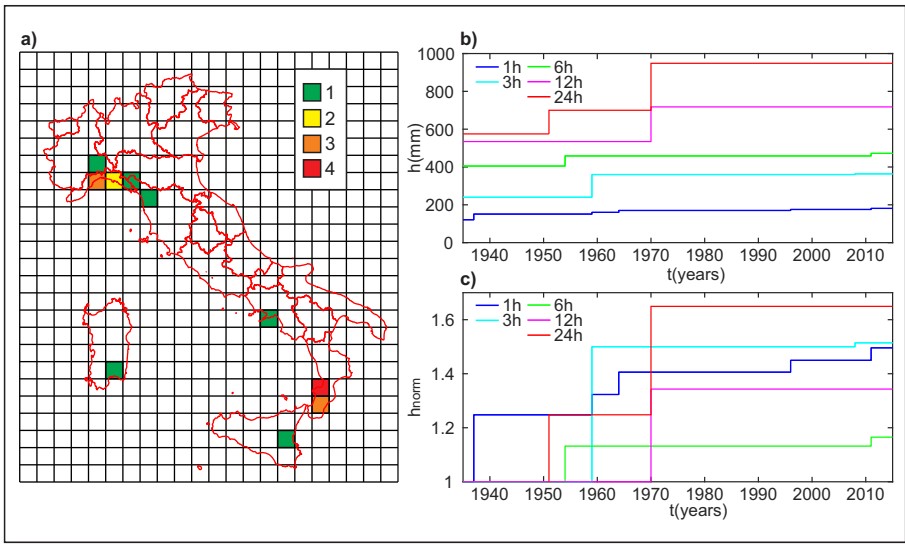

**Figure 5.** (a) Number of record-breaking events per cell over a 50 km grid. Record breaking rainfall depths for the 5 considered durations from 1935 to 2015 (b) in absolute values and (c) normalized on the 1935 values.

et al., 2014), due to the large recorded intensities and to their severe consequences, but the fragmented configurations of the national database have prevented a systematic treatment of this population of "extremes of the extremes" (Snorrason et al., 2002). A preliminary investigation on the occurrence of very-extreme events at the national scale has been performed and summarized in Figure 5a that shows the spatial distribution of the record-breaking rainfall events for the considered durations

from 1935 to 2015. A record-breaking event is defined as the annual value that exceeds all the previous ones. At this stage, only nationwide record-breaking are considered, pulling up all the data together year by year. Record breaking rainfall amounts can provide a picture of the spatio-temporal distribution of the major weather anomalies in the country. Analysing record-breaking events has some advantages from both an operational and a statistical point of view. Due to the significant amounts recorded, these events can be easily verified combining different sources of information, and, moreover, this kind of analysis does not

require any assumption on the underlying probability distribution (Coumou et al., 2013). The spatial distribution of the events seems to suggest a clusterization of these phenomena in some areas of the country: the eastern part of Liguria and northern part of Toscana and the extreme south of Calabria. Localized events also occurred in Campania, Sicilia and Sardegna. All of these areas are generally characterized by complex orography in the proximity of the coastline: a framework that can promote the development of particularly intense phenomena (Furcolo et al., 2015). At-site systematic analysis of the record breaking events,

which are expected to provide useful information for characterizing the extreme rainfall regime in the country (Lehmann et al., 2015), is now possible thanks to the consistency of the new *I-RED* database. Figure 5b shows the record-breaking evolution in time (for each duration); the occurrence of a new record-breaking is represented by an increasing step in the line. Figure 5c reports the same record-breakings, whose values are normalized by the 1935 values.

# 5 Conclusions

The first comprehensive dataset of extreme rainfall in Italy, called *I-RED*, has been presented here. It is a significant source of information, able to provide unprecedented knowledge on the characteristics of heavy precipitation in Italy and on the possible rainfall regime changes in the last century. Further efforts will be addressed to increase the spatial data homogeneity and coverage in time, by including the data of the most recent years and, eventually, by contacting the local authorities for requesting assistance in the merging of the series. The final aim is to make the update of the database systematic and unsupervised. This can be obtained by strengthening the collaboration with the data providers, in the framework of joined projects, as the one that led to the development of the *ArCIS* (Pavan et al., 2013) dataset, collecting updated rainfall and temperature data from a group of regional authorities in Northern Italy. Collaborations with other projects, focused on different spatial or temporal scales, will be also explored in order to automatically and efficiently analyse the consistency of the *I-RED* dataset and to integrate it with the existing ones. A possible target is the *SCIA* dataset (Desiato et al., 2007) referring to the 24-hour and daily scale. Joined projects with international institutions will be evaluated and endorsed in order to make available the *I-RED* database in larger frameworks for trans-boundary exchange of precipitation data. In the meanwhile the *I-RED* will be used for exploring the different outcomes provided by this preliminary analysis, e.g., assessing the influence of the spatial distribution of the stations on the observation of record-breaking extreme events, evaluating the presence of trends in the temporal distribution of the "black swans" and analysing the statistical predictability of these kind of events on such a wide and complex domain.

*Data availability.* The original data can be requested to the authorities reported in Table 1. Some of the agreements signed with the data providers, aimed at monitoring the correct use of the data, restrict their use to the aims of the authors' project. Due to these legal restrictions, the full or partial access to the *I-RED* can be provided:

- to research individuals or groups in the framework of the authors' project;

- to research individuals or groups not collaborating with the authors' project, upon evidence of permission received by the involved regional agencies, reported in Table 1.

For further details and queries, please refer to the corresponding author.

## Appendix A: Guidelines for the quality check of hydro-meteorological data

*Extracted and translated in English from Barbero S. et al. (2017)*

## A1 Quality control

[. . . ] The attribution of a certain level of quality to the measured data passes through the process of validation of the data themselves, which consists in analysing all the data collected in terms of completeness, reasonableness and in eliminating erroneous values. Data validation (validity check) is only one of the Quality Control operational procedures (QC) consisting of a set of

procedures and rules to ensure that a measurement system achieves and maintains a specific quality level initially established. The periodic calibration of the instruments, the periodic inspection of the sites and the preventive maintenance also belong to the QC process. The QC can be applied both in real time (real time quality control) that in delayed time, according to the needs of sharing, using and storing data nationally and internationally (e.g., World Meteorological Organization, 1993, 2010a, b).

Moreover, the QC must be integrated into an effective and well-coordinated Quality Management (QM). The QM is, in fact, is expressed through the joint application of Quality Assurance (QA) and of the QC (World Meteorological Organization, 2014). The QA is the set of planned and systematics activities applied within a quality management system to provide the level of confidence with which the quality requirements are met. Basically, the QC is a system of activities to provide a quality product while the QA is a system of activity designed to verify that the quality control system is functioning properly. The main objec-

tives of the QM are the identification, quantification and reduction of errors. Errors can be made for both technical reasons (i.e., due to the used methods and technologies) and procedural reasons (i.e., linked to an unclear or ineffective management or to the lack of adequate preparation of the operators). Furthermore, errors can be made during detection (e.g., the sensor does not read correctly), when the observation is transcribed (transposition of digits, shifting of dates, etc.) and during data transmission and storage (computer errors, errors of digitization, etc.). Many of these errors can be prevented by an appropriate QA, others

must be identified and corrected through the QC procedures.

## A2    Levels of the validity check

The first level of data validation is performed on the raw data (or gross data), i.e., the data at the original temporal resolution with which they are transmitted or detected at the measuring station and consists in the application of basic procedures for verifying the validity of the data. These checks aim at indicating malfunctions, instability, or interference. In the case of data

coming from automatic measuring stations the validity checks are applied to the "meteorological message" coming from the station in the transcoding phase of the message that for the transmission must comply with certain rules. The checks carried out will therefore be related to the expected formats within a given message, to the date and time stamps, to the location of measuring station, to the codes of stations and sensors and to the presence of duplicate elements. This category of checks includes: syntax controls (e.g., alphabetic characters appearing in a text that should be numeric) which, if incorrect, can mine the

transcoding process; logical controls that refer to both the intrinsic characteristics of the magnitude (e.g., World Meteorological Organization, 1993) and to the limits imposed by technical characteristics of the instrument used, in terms of measuring range (e.g., for a rain gauge: 0-300 mm/h), resolution (e.g., for a rain gauge: 0.1 or 0.2 mm), and limits in the operating temperatures (e.g., 0-70°C for an unheated or -30-70°C for a heated rain gauge). The first level of controls can have three types of outcomes: control passed, suspected data or control not passed. In the first two cases the data is then subjected to subsequent checks, in

the third the data is considered incorrect and discarded. The first level controls are performed on the elementary data, i.e., the data to the temporal aggregation derived from the measurement station. At this stage it is appropriate also carry out internal consistency checks. These are checks that are based on the comparison between synchronous values of different variables somehow related (e.g., by physical laws), so as to highlight any inconsistencies between the data. The second level of data validation consists of a series of "consistency" checks:

- Time consistency checks: they are based on the verification of the respect of a maximum and minimum level of variability of data over time and they have the purpose of identifying any anomalies between data temporally contiguous or with respect to the values that have historically occurred in a given site. Concerning the allowed minimum variability, consistency verification procedures are aimed at ascertaining the presence of persistence of measured values in the series, consisting in the lasting over time of a same or similar value;

- Cross checks with other quantities recorded at the same station: they are based on the control of the considered data with reference to other related quantities measured at the same site. E.g., temperature comparison with solar radiation;

- Spatial consistency checks: they are based on the hypothesis of gradual variability of the observed quantity in space and therefore on the existence of a sort of spatial correlation between the contemporaneous measures carried out in neighbouring stations. However, when dealing with rainfall, the hypothesis of gradual variability is lesser acceptable when smaller temporal aggregations are considered;

- Climatological checks: they are based on comparing the quantity under examination with some parameters derived from the whole historical series (e.g., tests based on the comparison with percentiles calculated on specific time intervals). The data are validated, at first, using automatic procedures. However, for the evaluation of the so-called "suspicious" data, a manual revision by qualified personnel is required to decide for every case if validate the suspect data, reject it as not valid, or fix it if possible.

*Competing interests.* No competing interests are present.

*Acknowledgements.* The authors thank Prof. Enrica Caporali and Dr. Valentina Chiarello for their assistance in preparing and screening the Toscana regional dataset, Dr. Stefano Macchia for his contribution in collecting and cleaning the data and the insightful comments of Prof. Alberto Montanari, three anonymous reviewers and the handling editor that allowed to significantly improve the quality of the original manuscript. Data providers reported in Table 1 are acknowledged.

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
