# Peer review of "Technical note: Space-time analysis of rainfall extremes in Italy: clues from a reconciled dataset"

_Hydrology and Earth System Sciences, 2017_

## Referee Comment (RC1) · Anonymous Referee #1 · 21 Jan 2018

The manuscript presents the Italian Rainfall Extreme Dataset (I-RED) of annual maxima values for 1-3-6-12 and 24 hours duration together with an exploratory statistical analysis of Italian annual maxima. The authors report in detail: (i) the efforts needed to collect the datasets from the complex universe of Italian institutions that are responsible for managing the measurement network; (ii) the techniques used and the choices made to merge the datasets into I-RED. Standard statistical techniques have been used for the statistical analysis and, as far as I can judge, without major flaws.

The presentation of the manuscript is clear and concise. The authors credit the sources of the data (even for those 12 Institutions out of 21 which have not yet provided data,

see Table 1, "Under request" under the column "Digitized data availability").

The potential of I-RED for both research and risk management/reduction is clearly evident and it is a pity that it cannot be made freely available because of the data policies of some of the institutions involved (as written in the paragraph "Data availability"). There are some questions I was left with after having gone through the manuscript about I-RED data policy and availability. In my opinion, this is the main limitation of your technical note. A little more detail on this major concern is below.

My advice to the editor is to accept the manuscript for publication with only minor revisions.

Major Comments:

- Data policy 1. The I-RED data policy is not clearly specified while it must be clearly presented to the reader both in the main text and briefly mentioned in the abstract too. The authors' work is valuable for the research community and the society whether I-RED could be made totally public or not. However, I believe most readers would be interested in knowing if I-RED is accessible in total or only in part. For instance, could you add a column to Table 1 to make it clear if you are allowed to re-distribute the data through I-RED? Is there any website where the reader could access the public part of I-RED?

- Data policy 2. You made clear in the manuscript that I-RED is related to CUBIST, which can be accessed on the polito website and as you write it is "first important attempt of making the large Italian hydrological heritage freely available in computer-readable format" (page 2, line 19). My question is: why CUBIST data is freely available and I-RED data has restricted access?

- Data policy 3. How could you reconcile these two statement: (1) page 2 line 27 "...the Italian law adopted an Open Source policy for the public data..." and (2) your statement on "data availability" where you write that you have signed

agreements with the data provider (which are all public institutions) that restrict the use of the data to the aims of your project? In my view, if you get the dataset only after signing an agreement that limit the use of a dataset then the dataset is not openly available (by definition).

- Conclusions and Future plans. Page 10, line 14. "The final aim is to make the update of the database systematic and unsupervised." A few words on how the authors plan to achieve the goal would do here. In particular, if the author could present any link with national and/or international project /ctivities this would strengthen their statement. For example, I'm aware of initiatives aimed at collecting Italian datasets such as ArCIS (https://www.arcis.it/wp/en/home-2/) or ISPRA-SCIA (http://www.scia.isprambiente.it/home_new.asp). Do the authors have any contact with them? In particular, it seems to me that ISPRA-SCIA is doing a work on the establishment of a national database that is very similar to I-RED, though mostly for aggregated values. Do you plan to join your efforts with other institutions? Do you have any contact with international institutions? There are several ongoing projects at an international level to collect and organize in-situ observations (See for example COPERNICUS: https://climate.copernicus.eu/global-land-and-marine-observations-database)

Minor Comments:

- Figure 2. Panels b and c shows exactly the same histogram. Please, check it.

- Figure 4. Excellent Figure. Page 8, lines 7-8 ("When short...appears."). I agree with your statement, nonetheless this could also be an artifact due to the color scale chosen. Is there any particular reason for your choices of min/max values for color scale of the different durations? Does the min/max values correspond to any percentile of the distribution of values, for example?

[Figure]

---

## Author Comment (AC1) · 4 Feb 2018

We greatly appreciate the insightful comments from the reviewer. The comments from the reviewer have been reproduced in italic below, interspersed with our responses.

*The manuscript presents the Italian Rainfall Extreme Dataset (I-RED) of annual maxima values for 1-3-6-12 and 24 hours duration together with an exploratory statistical analysis of Italian annual maxima. The authors report in detail: (i) the efforts needed to collect the datasets from the complex universe of Italian institutions that are responsible for managing the measurement network; (ii) the techniques used and the choices made to merge the datasets into I-RED. Standard statistical techniques have been*

*used for the statistical analysis and, as far as I can judge, without major flaws. The presentation of the manuscript is clear and concise. The authors credit the sources of the data (even for those 12 Institutions out of 21 which have not yet provided data, see Table 1, "Under request" under the column "Digitized data availability").*

We need to clarify a misunderstanding on this point. In Table 1, the "Under request" category is related to those agencies that do not provide data directly on their websites. Most of them have already provided their datasets (or a part of them) upon request. As it is not clear, the Table will be edited in the revised manuscript under the category "available upon request".

*The potential of I-RED for both research and risk management/reduction is clearly evident and it is a pity that it cannot be made freely available because of the data policies of some of the institutions involved (as written in the paragraph "Data availability"). There are some questions I was left with after having gone through the manuscript about I-RED data policy and availability. In my opinion, this is the main limitation of your technical note. A little more detail on this major concern is below. My advice to the editor is to accept the manuscript for publication with only minor revisions.*

*Major Comments:*

*1. Data policy 1. The I-RED data policy is not clearly specified while it must be clearly presented to the reader both in the main text and briefly mentioned in the abstract too. The authors' work is valuable for the research community and the society whether I-RED could be made totally public or not. However, I believe most readers would be interested in knowing if I-RED is accessible in total or only in part. For instance, could you add a column to Table 1 to make it clear if you are allowed to re-distribute the data through I-RED? Is there any website where the reader could access the public part of I-RED?*

We thank the reviewer for the suggestion. Data availability is one of our main concerns; that is why we have reported in the caption of Table 1 all the websites where the reader

can find the original datasets. In the revised version of the manuscript we plan to add information on the procedure to access the data that we have merged and harmonized in the *I-RED*. In essence, as the data owners want to supervise their use, we only have permission to use them for purposes connected to our project. Consequently, we can provide the full database access only to research individuals or groups who join our project. Nevertheless, some individual regional databases can be provided upon evidence of permission received by the regional agencies, in particular, those that release data upon request.

*2. Data policy 2. You made clear in the manuscript that I-RED is related to CUBIST, which can be accessed on the polito website and as you write it is "first important attempt of making the large Italian hydrological heritage freely available in computer-readable format" (page 2, line 19). My question is: why CUBIST data is freely available and I-RED data has restricted access?*

This is a drawback of the change in the owner of the network that we have described in the manuscript. The data included in the *CUBIST* database are the digitized version of the data included in the Hydrological Yearbooks of the *SIN-SIMN*, that are freely available in a PDF format under the Hydrological Yearbooks Project of the Italian National Institute for Environmental Protection and Research (*ISPRA*) (http://www.isprambiente.gov.it/it/progetti/acque-interne-e-marino-costiere-1/progetto-annali). As a digital version of public data, they are public too. After the *SIMN* has been dismantled, every local authority has adopted a different policy for data distribution. This further clarifies the difficulties we have in making the *I-RED* dataset fully open. Moreover, the data are often provided only upon request, because the regional authorities want to supervise the spreading and the correct use of them, as stated in answer 1.

*3. Data policy 3. How could you reconcile these two statement: (1) page 2 line 27 "...the Italian law adopted an Open Source policy for the public data..." and (2) your statement on "data availability" where you write that you have signed agreements with*

[Figure]

*the data provider (which are all public institutions) that restrict the use of the data to the aims of your project? In my view, if you get the dataset only after signing an agreement that limit the use of a dataset then the dataset is not openly available (by definition).*

The reviewer is right: the two sentences seem to be incompatible, but the situation is the result of the evolving framework of the national and regional laws concerning the "open data policy". The italian law concerning the open data refers mainly to the D.Lgs.82/2005, D.Lgs. 36/2006, Law 221/2012, D.Lgs. 179/2012, Law 114/2014. Substantially it requires that all the data of public interest collected from public authorities should be made freely available on the internet for non-commercial use. The complex framework of the regional hydrological agencies actually comply to the law in a varied fashion. Some of the regional agencies started immediately to provide the data on the internet as they become available. Other agencies, to make sure of the non-commercial use of the data, provide them only upon request, after a certification of the purposes for which the data would be used. Other regions are still developing the platform for the digitalization and the distribution of the data, and their local archive are still not officially ready for the publication. Moreover, in some regions agencies became full operational after the dismissal of the *SIMN* only years later: in this time lag data may have been collected in a non-systematic way from authorities different from the one which finally came in charge of the service. This involves further complexity in the definition of who and how should work on patching and publishing the data. Considering the global complexity of the topic, supplementary material including the official data policies of the different authorities will be added.

*4. Conclusions and Future plans. Page 10, line 14. "The final aim is to make the update of the database systematic and unsupervised." A few words on how the authors plan to achieve the goal would do here. In particular, if the author could present any link with national and/or international project/activities this would strengthen their statement. For example, I'm aware of initiatives aimed at collecting Italian datasets such as ArCIS (https://www.arcis.it/wp/en/home-2/) or ISPRA-SCIA (http://www.scia.isprambiente.it).*

*Do the authors have any contact with them? In particular, it seems to me that ISPRA-SCIA is doing a work on the establishment of a national database that is very similar to I-RED, though mostly for aggregated values. Do you plan to join your efforts with other institutions? Do you have any contact with international institutions? There are several ongoing projects at an international level to collect and organize in-situ observations (See for example COPERNICUS: https://climate.copernicus.eu/global-land-and-marine-observations-database)*

The reviewer is right. There are other significant projects in the Italian framework that will be mentioned in the revised version of the manuscript. In particular, the *ISPRA-SCIA* dataset pursues the same aims of this work, but just relative to the daily precipitation. The *ISPRA-SCIA* dataset is a significant source of data, considered also in the development of the *CUBIST* database and a mention to the project will be added in the revised version of the manuscript. In general, the projects the reviewer refers to (as most of the project related to hydrology in Italy) have been developed from the same agencies that collaborated with us for the development of the *I-RED*, as partners of *ISPRA*. We plan to contact again all the agencies for a feedback on the work that we have done, in order to strength the collaborations and integrate our work in a national framework. We are also looking forward to join to international institutions in starting EU-wide projects aimed at maximizing the outcomes of this work.

*Minor Comments:*

*4. Figure 2. Panels b and c shows exactly the same histogram. Please, check it.*

The distribution of the length classes is similar across the different durations. As it is not possible nor useful to show such behavior, figure 2 will be edited in the revised manuscript.

*5. Figure 4. Excellent Figure. Page 8, lines 7-8 ("When short...appears."). I agree with your statement, nonetheless this could also be an artifact due to the color scale chosen. Is there any particular reason for your choices of min/max values for color*

*scale of the different durations? Does the min/max values correspond to any percentile of the distribution of values, for example?*

The reviewer is right, the apparent distribution of the median values could be also an artifact of the chosen color scale (the min/max values correspond to the min/max values of the distributions). Nonetheless, the coefficient of variation of the nationwide distribution of the medians grows when larger durations are considered, and this seems to confirm our hypothesis on more "peaky" distribution of larger durations values. These considerations are included in the revised manuscript.

---

## Referee Comment (RC2) · Anonymous Referee #2 · 5 Feb 2018

This technical note presents a unified database of precipitation extremes over Italy. There is no doubt that such efforts aiming to gather, "clean up" and finally provide all the available information in "one place" are very useful for hydrological design. Of course in this specific case these efforts are undermined by the fact that the final database is not actually freely available. In the open-access era this is a serious drawback which however, if I understood correctly, it is not authors' fault but a restriction for the Italian authorities. There are a few minor comments and suggestions that I would like to make hoping to be helpful in improving this technical note. 1. Figure 2: There is not any official abbreviation of years as "Y" so probably it would be more clear instead of t(Y) to write just "Years". Panels b and c: please decrease the size of fonts in the

[Figure]

X-axis so it can be read more easily and also change the label to "Length (years)" or something similar as no.years is confusing (also check the panel d: do you mean years or data?). This suggestion is for any other Figure, e.g., for Figure 5 where "t(Y)" is used. 2. Analysis presented in Figure 4. There is a dense network, more than sufficient to provide kriging estimates for the whole Italy. It would much more useful in my opinion not only because dots may overlap but because you will provide estimates also in places where there is no information. So, I would suggest to construct kriging maps of the statistics analyzed. 3. Mean is quite robust in general, yet here you prefer only the median. Of course it can be affected by outliers yet common methods of fitting distributions, e.g., product moments or L-moments are using mean values. So, in my opinion you should provide also the maps of the mean value. 4. If I understood well you have estimated the mean values of L-CV, L-skew and L-kurt of all duration. Of course you are dealing with maxima and we are expecting the shape characteristics to be close yet this is not necessarily true. If indeed these summary statistics are close among the different duration, please report it or else provide different maps for each duration. 5. It is not clear to me if the maxima values have emerged from a sliding-window process or from a fixed-block (non-overlapping). In the latter case the user of this database should know this fact in order to correct the data by the Hershfield factor. Please comment on that and clarify. 6. You can use plain text for L-CV. In Figure 4 is plain while in the text you are using Italics. Please re-check the text for minor typos, e.g., line 26, p6 replace "an unique" with "a unique".

---

## Referee Comment (RC3) · A. Montanari (Referee) · 6 Feb 2018

The paper presents a data-set of extreme rainfall in Italy. I believe that the information that is provided here is potentially very interesting. An important question today is whether short duration precipitation has been impacted by climate change. Extreme rainfall with sub-hourly duration is relevant for the generation of flash-flood events that are a reason of concern for small to medium size catchments, which are numerous in the Alpine region. Flash floods recently caused several deadly events in Italy whose frequency is markedly increasing in recent times, therefore pointing out the need for mitigation strategies. These latter need to be designed basing on updated information

on extreme rainfall, with sub-hourly duration, that is rarely available. For this reason, I found this paper extremely interesting as it provides an example to follow and paves the way for elaborating and addressing very important research questions.

I believe the paper is well written and organized. I have minor suggestions to forward to the authors.

1) Abstract: I would avoid the term "explosive rainfall". I understand the reason why the authors introduced it in the first sentence of the abstract, but I still believe that it would be advisable to use terms with a well defined technical meaning.

2) Table 1: I do not understand the meaning of "under request". Does this mean that data are not yet available? Were the data already requested? Figure 2 shows that information for some of the regions labeled as "under request" is already available and therefore the whole picture is not completely clear to me.

3) Page 3, line 11: it is stated that "Considering that most of the provided data have been validated from the related authorities, they are considered reliable and, at first, included directly in the I-RED." It would be interesting to discuss the validation tests that have been considered by the authorities. Extreme rainfall data may be affected by relevant uncertainty, it would be useful to mention the gauging methods, what kind of checks have been considered by the authorities and so on.

4) Page 9, line 14: "A record-breaking event is defined as the annual value that exceeds all the previous ones." Such definition implies a greater frequency of events at the beginning of the record. Did the authors consider identifying record-breaking events by fixing a threshold for rainfall intensity, basing on information that may be extracted by the whole record of observations (without introducing any assumption on the underlying probability distribution?

5) Page 9, line 14: it is stated that "At this stage, only nationwide record-breakings are considered, pulling up all the data together year by year." I do not understand how

data were pulled together. Did the authors pool together sites with different climatic behaviours? Does this mean that record-breaking events at the local level may have been discarded?

6) Data availability is a potential issue. Data-bases are useful when they are readily available. It would be interesting to discuss data availability in the body of the paper.

Overall, I am strongly in favor of publication. I believe this paper may pave the way for setting up transboundary initiatives for putting together extended information on extreme rainfall. Such data would provide an essential information for better understanding flash floods and climate change.

---

## Referee Comment (RC4) · Anonymous Referee #4 · 8 Feb 2018

This technical note describes a unique extreme rainfall dataset (I-RED) that was compiled for Italy. The introduction nicely explains the complexities of the rainfall data records in Italy and the necessity for a unified dataset. The labor involved in creating such a dataset is appreciated and will be valuable for future research. The I-RED compilation methods is described relatively well, and some initial results are briefly discussed; however, I believe minor revisions are necessary to close some open questions that a reader may have about the dataset. Specific comments are below.

1) Some discussion is needed on how the rainfall data at these stations is collected, as well as how the practice of recording rainfall data over time may have changed.

[Figure]

For example, there may be some stations that reported 24 h rainfall totals for a longer period of time than 1 h rainfall recordings. This additional discussion could be aided by a plot similar to Figure 2d for the shortest (1 h) and longest (24 h) durations. Information on the types of gauges at these stations could be critical to understanding how capable the dataset is at capturing the most extreme rainfall amounts.

2) Page 3 Lines 14-15: How frequently were data types (b) and (c) inconsistent with one another, and what would be the source of this inconsistency if it's the same station?

3) Figure 2 (b) and (c): These histograms look identical. Is it true that the same number of stations that reported 24 h rainfall also reported 1 h rainfall?

4) Table 1: It is unclear what "under request" means here. Does it mean "available upon request"?

5) There are a few typos in the paper, including the following: Page 1 Line 24: "dismantledment" should be "dismantlement", Page 1 Line 25: "stucked" should be "stuck", Page 2 Line 2: "metodologies" should be "methodologies".

---

## Author Comment (AC2) · 21 Feb 2018

We greatly appreciate the insightful comments from the reviewer. The comments from the reviewer have been reproduced in italic below, interspersed with our responses.

*This technical note presents a unified database of precipitation extremes over Italy. There is no doubt that such efforts aiming to gather, "clean up" and finally provide all the available information in "one place" are very useful for hydrological design. Of course in this specific case these efforts are undermined by the fact that the final database is not actually freely available. In the open-access era this is a serious drawback which however, if I understood correctly, it is not authors' fault but a restriction for the Italian*

*authorities.*

The reviewer is right. As we have previously stated in the answers to the reviewer 1 open data availability is one of our main concerns. However, this only affects the fact that data are not all available in one place. In essence, as the data owners want to supervise their use and we have the permission to use them for purposes connected to our project, we can provide the full database access only to research individuals or groups who join our project. Nevertheless, we can redistribute most regional databases upon evidence of permission received by the relevant agencies. Taking into account these limitations, clarifications on the procedure to access the data that we have merged and harmonized in the I-RED will be added in the revised version of the manuscript.

There are a few minor comments and suggestions that I would like to make hoping to be helpful in improving this technical note.

***Major Comments:***

*1. Figure 2: There is not any official abbreviation of years as "Y" so probably it would be more clear instead of t(Y) to write just "Years". Panels b and c: please decrease the size of fonts in the X-axis so it can be read more easily and also change the label to "Length (years)" or something similar as no.years is confusing (also check the panel d: do you mean years or data?). This suggestion is for any other Figure, e.g., for Figure 5 where "t(Y)" is used.*

The figures will be corrected in the revised version of the manuscript, according to the suggestion of the reviewer.

*2. Analysis presented in Figure 4. There is a dense network, more than sufficient to provide kriging estimates for the whole Italy. It would much more useful in my opinion not only because dots may overlap but because you will provide estimates also in places where there is no information. So, I would suggest to construct kriging maps of*

*the statistics analyzed.*

Considering the exploratory nature of our analysis we did prefer to provide point statistics, which –at this stage- indicate the main features of rain variability at the country scale but do not address the goal of providing robust estimates in ungauged areas. This is a very important research objective to address. However, considering the many problems we have faced in the regional analysis in the North-West of Italy (Libertino et al, 2018) the uneven distribution of the rain gauges in space and the high variability of record length make the whole kriging analysis a challenge deserving a separate work. In partially following the reviewer's suggestion, however, interpolated maps will be added as supplementary material to more clearly represent rainfall indices variability over space.

*3. Mean is quite robust in general, yet here you prefer only the median. Of course, it can be affected by outliers yet common methods of fitting distributions, e.g., product moments or L-moments are using mean values. So, in my opinion you should provide also the maps of the mean value.*

The reviewer is right: most of the common methods of fitting distributions consider the mean as best descriptor of the central tendency. However, in this descriptive phase we wanted to represent the variability of the central tendency of the series as unbiased as possible with respect to the outliers. On the other hand, we admit information of the mean can be of interest for the readers: therefore, we will consider to add a map of the mean of the extremes it in the revised version of the manuscript.

*4. If I understood well you have estimated the mean values of L-CV, L-skew and L-kurt of all duration. Of course you are dealing with maxima and we are expecting the shape characteristics to be close yet this is not necessarily true. If indeed these summary statistics are close among the different duration, please report it or else provide different maps for each duration.*

The reviewer is right. Considerations on the behaviour of these statistics across the

different durations will be added in the revised version of the manuscript.

*5. It is not clear to me if the maxima values have emerged from a sliding-window process or from a fixed-block (non-overlapping). In the latter case the user of this database should know this fact in order to correct the data by the Hershfield factor. Please comment on that and clarify. Annual maxima have been collected with a sliding-windows procedure. The information will be added in the revised version of the manuscript.*

6. You can use plain text for L-CV. In Figure 4 is plain while in the text you are using Italics. Please re-check the text for minor typos, e.g., line 26, p6 replace "an unique" with "a unique".

We thank the reviewer for the corrections. The manuscript will be double-checked for typos.

**References**

Libertino A., P. Allamano, F. Laio, and P. Claps. Regional-scale analysis of extreme precipitation from short and fragmented records. Advances in Water Resources 2018, 112, 147-159, doi: 10.1016/j.advwatres.2017.12.015

---

## Author Comment (AC3) · 21 Feb 2018

We greatly appreciate the insightful comments from the reviewer. The comments from the reviewer have been reproduced in italic below, interspersed with our responses.

*The paper presents a data-set of extreme rainfall in Italy. I believe that the information that is provided here is potentially very interesting. An important question today is whether short duration precipitation has been impacted by climate change. Extreme rainfall with sub-hourly duration is relevant for the generation of flash-flood events that are a reason of concern for small to medium size catchments, which are numerous in the Alpine region. Flash floods recently caused several deadly events in Italy whose*

*frequency is markedly increasing in recent times, therefore pointing out the need for mitigation strategies. These latter need to be designed basing on updated information on extreme rainfall, with sub-hourly duration, that is rarely available. For this reason, I found this paper extremely interesting as it provides an example to follow and paves the way for elaborating and addressing very important research questions. I believe the paper is well written and organized. I have minor suggestions to forward to the authors.*

*1. Abstract: I would avoid the term "explosive rainfall". I understand the reason why the authors introduced it in the first sentence of the abstract, but I still believe that it would be advisable to use terms with a well-defined technical meaning.*

We have used the term "explosive rainfall", recently become popular in the media and in the common language, to more easily reach a wider audience. The term is not commonly adopted by the average scientific community to which this journal is addressed, while meteorologists and climatologists are more incline to relate heavy thunderstorms to explosive cyclones (see e.g. Koroutzoglou et al, 2011) and this motivates the adoption of the term. However, we accept the suggestion to refer to a more technically sound term, and we will try to clarify with a separate sentence and with simple words the significance of the phenomena under study.

*2. Table 1: I do not understand the meaning of "under request". Does this mean that data are not yet available? Were the data already requested? Figure 2 shows that information for some of the regions labeled as "under request" is already available and therefore the whole picture is not completely clear to me.*

As we explained in the answers to reviewer 1 and 4, we were referring to those cases in which the agencies do not provide data directly on their websites but they provide it "upon request". In the revised manuscript the "Under request" wording will be substituted by the more explicit "available upon request".

*3. Page 3, line 11: it is stated that "Considering that most of the provided data have*

*been validated from the related authorities, they are considered reliable and, at first, included directly in the I-RED"* It would be interesting to discuss the validation tests that have been considered by the authorities. Extreme rainfall data may be affected by relevant uncertainty, it would be useful to mention the gauging methods, what kind of checks have been considered by the authorities and so on.

The reviewer is right. We can not provide fully detailed information on the measurement methods and on the validation, which are certainly under the WMO standards but not necessarily free of possible instruments inaccuracies. This detailed information is not generally provided by the surveying authorities. Part of the data are from manual gauges and most of them are from automatic gauges, but this basic kind of information is also lacking. We will further contact the authorities and/or explore their websites for investigating the existence of additional published or unpublished works providing information on the gauging and validating operations. Such information will be eventually translated (most of them are in Italian) and summarized as supplementary material.

*4. Page 9, line 14: "A record-breaking event is defined as the annual value that exceeds all the previous ones."* Such definition implies a greater frequency of events at the beginning of the record. Did the authors consider identifying record-breaking events by fixing a threshold for rainfall intensity, basing on information that may be extracted by the whole record of observations (without introducing any assumption on the underlying probability distribution. *5. Page 9, line 14: it is stated that "At this stage, only nationwide record-breakings are considered, pulling up all the data together year by year."* I do not understand how data were pulled together. Did the authors pool together sites with different climatic behaviours? Does this mean that record-breaking events at the local level may have been discarded?

At this stage we provide a unique answer to the two questions, as they deal with the same topic. We are aware that to carry on analysis that involves pooling up of the data requires homogeneity controls and the wide domain under analysis does not warrant for climatic homogeneity. However, we propose a countrywide record-breaking analysis

to provide a general assessment on the large rainfall amounts we are referring to when speaking about the "extremes of the extremes". Location of the record breaking also point out their distribution in space. This analysis does not allow to consider record breaking events at the local level. We will clarify this in the revised version of the manuscript. On the other hand, our research on the topic is ongoing, as we are trying to identify a meaningful threshold for selection of homogeneous extraordinary events at the country scale. This additional analysis is still under development and will require a significant amount of efforts. We have then decided to devote a different paper to this analysis, that can have implication in terms of significance of increasing trends of short duration rainstorms.

*6. Data availability is a potential issue. Data-bases are useful when they are readily available. It would be interesting to discuss data availability in the body of the paper.*

The reviewer is right. As we have previously declared in the answers to the other reviewers, data availability is one of our main concern. In essence, as the data owners want to supervise their use, we only have permission to use them for purposes connected to our project. Consequently, we can provide the full database access only to research individuals or groups who join our project. Nevertheless, some individual regional databases can be provided upon evidence of permission received by the regional agencies, in particular, those that release data upon request. Taking into account these limitations, clarifications on the procedure to access the data that we have merged and harmonized in the I-RED will be added in the revised version of the manuscript and additional information will be attached in the supplementary material.

*Overall, I am strongly in favor of publication. I believe this paper may pave the way for setting up transboundary initiatives for putting together extended information on extreme rainfall. Such data would provide an essential information for better understanding flash floods and climate change.*

**References**

Kouroutzoglou, J., Flocas, H. A., Keay, K., Simmonds, I., Hatzaki, M. (2011). Climatological aspects of explosive cyclones in the Mediterranean. International Journal of Climatology, 31(12), 1785-1802.

---

## Author Comment (AC4) · 21 Feb 2018

We greatly appreciate the insightful comments from the reviewer. The comments from the reviewer have been reproduced in italic below, interspersed with our responses.

*This technical note describes a unique extreme rainfall dataset (I-RED) that was compiled for Italy. The introduction nicely explains the complexities of the rainfall data records in Italy and the necessity for a unified dataset. The labor involved in creating such a dataset is appreciated and will be valuable for future research. The I-RED compilation methods is described relatively well, and some initial results are briefly discussed; however, I believe minor revisions are necessary to close some open ques*

[Figure]

*tions that a reader may have about the dataset. Specific comments are below.*

*1. Some discussion is needed on how the rainfall data at these stations is collected, as well as how the practice of recording rainfall data over time may have changed. For example, there may be some stations that reported 24 h rainfall totals for a longer period of time than 1 h rainfall recordings. This additional discussion could be aided by a plot similar to Figure 2d for the shortest (1 h) and longest (24 h) durations. Information on the types of gauges at these stations could be critical to understanding how capable the dataset is at capturing the most extreme rainfall amounts.*

*3. Figure 2 (b) and (c): These histograms look identical. Is it true that the same number of stations that reported 24 h rainfall also reported 1 h rainfall?*

At this stage we provide a unique answer to the two questions, as they deal with the same topic. In general, all the stations have had for the whole operating period a resolution in time amenable for capturing the maxima for all the considered durations. However, some data could have been discarded from the authorities for different reasons, that is why the number of data for the different duration can differ. These differences are quite small, that is why the two histograms in figures 2(b) and 2(c) look similar (see question). We will find a better way to underline these differences in the revised version of the manuscript. We will also add some considerations on the characteristics of the rain gauges.

*2. Page 3 Lines 14-15: How frequently were data types (b) and (c) inconsistent with one another, and what would be the source of this inconsistency if it's the same station?*

The sources of the inconsistencies could be various, according to the evolution of the monitoring systems of the different regions and often is due to the joint management of interregional basins. The different regional authorities often have adopted different codes/names for the same station, the first step has been thus to identify the presence of duplicate stations with same/similar name covering different time intervals. Sometimes, even for the same station, neighboring regions can provide different data for the

same years. This can be, e.g., due to the fact that sometimes regions share rainfall data before their validation and official publication. That is why we decided, in case of inconsistencies, to preserve the data from the competent authority.

*3. Table 1: It is unclear what "under request" means here. Does it mean "available upon request"?*

As we explained in the answers to reviewer 1 and Alberto Montanari, we were referring to those cases in which the agencies do not provide data directly on their websites but they provide it "upon request". In the revised manuscript the "Under request" wording will be substituted by the more explicit "available upon request".

*4. There are a few typos in the paper, including the following: Page 1 Line 24: "dis-mantledment" should be "dismantlement", Page 1 Line 25: "stucked" should be "stuck", Page 2 Line 2: "metodologies" should be "methodologies".*

We thank the reviewer for the corrections. The manuscript will be double-checked for typos.
* * *

---

## Author Response (AR1)

Technical note: Space-time analysis of rainfall extremes in Italy: clues from a reconciled dataset
Libertino, Andrea*; Ganora, Daniele; Claps, Pierluigi
*andrea.libertino@polito.it

We greatly appreciate the insightful comments from the editors and reviewers. The comments from the editors and reviewers have been reproduced below, interspersed with our indented responses (R#1, R#2 and R#4 refers to comments from the anonymous reviewers 1,2 and 4 respectively, R#AM are comment from the reviewer Alberto Montanari and E refers to Editor's comments). The comments have been grouped when dealing with similar topics. The line numbering refers to the numbering of the revised manuscript. To help identifying the updates made to the manuscript, a copy with the track changes is attached too.

**R#1_C0** *The presentation of the manuscript is clear and concise. The authors credit the sources of the data (even for those 12 Institutions out of 21 which have not yet provided data, see Table 1, "Under request" under the column "Digitized data availability")*
**R#AM_C2** *Table 1: I do not understand the meaning of "under request". Does this mean that data are not yet available? Were the data already requested? Figure 2 shows that information for some of the regions labeled as "under request" is already available and therefore the whole picture is not completely clear to me.*
**R#4_C4** *Table 1: It is unclear what "under request" means here. Does it mean "available upon request"?*

> We were referring to those cases in which the agencies do not provide data directly on their websites but they provide it "upon request". In the revised manuscript the "Under request" wording is substituted in Table 1 by the more explicit "available upon request".

**R#1_C1** *Data policy 1. The I-RED data policy is not clearly specified while it must be clearly presented to the reader both in the main text and briefly mentioned in the abstract too. The authors' work is valuable for the research community and the society whether I-RED could be made totally public or not. However, I believe most readers would be interested in knowing if I-RED is accessible in total or only in part. For instance, could you add a column to Table 1 to make it clear if you are allowed to re-distribute the data through I-RED? Is there any website where the reader could access the public part of I-RED?*
**R#1_C2** *Data policy 2. You made clear in the manuscript that I-RED is related to CUBIST, which can be accessed on the polito website and as you write it is "first important attempt of making the large Italian hydrological heritage freely available in computer-readable format" (page 2, line 19). My question is: why CUBIST data is freely available and I-RED data has restricted access?*
**R#1_C3** *Data policy 3. How could you reconcile these two statement: (1) page 2 line 27 "...the Italian law adopted an Open Source policy for the public data..." and (2) your statement on "data availability" where you write that you have signed agreements with the data provider (which are all public institutions) that restrict the use of the data to the aims of your project? In my view, if you get the dataset only after signing an agreement that limit the use of a dataset then the dataset is not openly available (by definition).*
**R#AM_C6** *Data availability is a potential issue. Data-bases are useful when they are readily available. It would be interesting to discuss data availability in the body of the paper.*
**E** *Please, explain into detail the availability of data (such as upon request), as this seems to be the major plus of the paper for a wide audience that there is a set of data available to many researchers not familiar with the data set in Italy and how to get the data.*

> We thank the reviewers and the editor for the suggestions. Data availability is one of our main concerns, that is why we have reported in the caption of Table 1 all the websites where the reader could find the original regional datasets. In the revised version of the manuscript have described the procedure to access the data that we have merged and harmonized in the *I-RED* in the "data availability" box. Considering the relevance of the topic, some lines about the accessibility of the *I-RED* have been also added in the abstract (P1, LL10-11) and in the manuscript (P7, LL4-5). Substantially, as the owners of the data want to supervise the use of their data and have restricted the access to the aim of our project, we can provide the full database only to those who will join our project and collaborate in its development. We plan also to provide the access to the regional databases to those who will provide us a clearance from the regional authorities in which they are interested.
>
> We understand that it is not easy to understand why the data of the *CUBIST* dataset is effectively open to everybody while the equivalent data from the local agencies have a restricted access, but this is mainly a drawback

of the change in the owner of the network that we have described in the manuscript. The data included in the CUBIST database are the digitized version of the data included in the Hydrological Yearbooks of the *SIN-SIMN* that have been made freely available in a PDF format under the Hydrological Yearbooks Project of the Italian National Institute for Environmental Protection and Research (*ISPRA*) (http://www.isprambiente.gov.it/it/progetti/acque-interne-e-marino-costiere-1/progetto-annali). As digital version of public data, they are public too. When the *SIMN* have been dismantled every local authority adopt a different policy for data providing and distribution. Generally, even if they are still available for research and no-profit purposes, they are monitored from the competent authorities. From this emerges the difficulties in making the *I-RED* dataset fully open access. The italian law concerning the open data mainly refers to the D.Lgs.82/2005, D.M.10/11/2011, D.Lgs. 36/2006, L. 221/2012, D.Lgs. 179/2012, L. 114/2014 as clarified at P2, L33. Substantially it requires that all the data of public interest collected from public authorities should be made freely available on the internet for non-commercial use. The complex framework of the hydrological authorities answered differentially to the law. Some of the regional authorities started immediately to provide the data openly on the internet as they become available. Other regions, to make sure of the non-commercial use of the data, provide them only upon request, after a certification of the purposes for which the data would be used. Other regions are still developing the platforms for the digitalization and the distribution of the data, and their local archives are still not officially ready for the publication. Moreover, there is a gap in some regions between the dismissal of the SIMN and the full operationality of the local authorities: in this period data could have been collected in a non-systematic way from authorities different from the definitive one. This involves further complexity in the definition of who and how should work on patching and publishing the data.

*R#1_C4* *Conclusions and Future plans. Page 10, line 14. "The final aim is to make the update of the database systematic and unsupervised." A few words on how the authors plan to achieve the goal would do here. In particular, if the author could present any link with national and/or international project/activities this would strengthen their statement. For example, I'm aware of initiatives aimed at collecting Italian datasets such as ArCIS (https://www.arcis.it/wp/en/home-2/) or ISPRA-SCIA (http://www.scia.isprambiente.it/home_new.asp). Do the authors have any contact with them? In particular, it seems to me that ISPRA-SCIA is doing a work on the establishment of a national database that is very similar to I-RED, though mostly for aggregated values. Do you plan to join your efforts with other institutions? Do you have any contact with international institutions? There are several ongoing projects at an international level to collect and organize in-situ observations (See for example COPERNICUS: https://climate.copernicus.eu/global-land-and-marine-observations-database)*

We have clarified the part of the manuscript related to the future plans. As clarified at P11, LL4-14 the automatic and unsupervised update of the database can be obtained by strengthening the collaboration with the data providers, in the framework of joined projects, as the one that led to the development of the *ArCIS* dataset, collecting updated rainfall and temperature data from a group of regional authorities in Northern Italy. The consistence with other projects working at other spatio-temporal scales will be also carried out in order to automatically analyse the consistence of the developed dataset and integrating it with the existing ones, e.g, the *SCIA* dataset, referring to the 24 hours and daily scale. Joined projects with international institutions will be evaluated and endorsed in order to make available the *I-RED* database in larger frameworks for trans-boundary exchange of precipitation data.

*R#4_C3* *Page 3 Lines 14-15: How frequently were data types (b) and (c) inconsistent with one another, and what would be the source of this inconsistency if it's the same station?*

As clarified at P5, L1-8 of the revised manuscript, the sources of the inconsistencies could be various, according to the evolution of the monitoring systems of the different regions and often is due to the joint management of interregional basins. The different regional authorities often have adopted different codes/names for the same station, the first step has been thus to identify the presence of duplicate stations with same/similar name covering different time intervals. Sometimes, even for the same station, neighboring regions can provide different data for the same years. This can be, e.g., due to the fact that sometimes regions share rainfall data before their validation and official publication. That is why we decided, in case of inconsistencies, to preserve the data from the competent authority.

*R#AM_C1* *Abstract: I would avoid the term "explosive rainfall". I understand the reason why the authors introduced it in the first sentence of the abstract, but I still believe that it would be advisable to use terms with a well-defined technical meaning.*

We have used the term "explosive rainfall", recently become popular in the media and in the common language, to more easily reach a wider audience. The term is not commonly adopted by the average scientific community to which this journal is addressed, while meteorologists and climatologists are more incline to relate heavy thunderstorms to explosive cyclones (see e.g. Kouroutzoglou et al, 2011) and this motivates the adoption of the term. However, we accept the suggestion to refer to a more technically sound term, as in P1, L1 of the revised manuscript.

*R#AM_C4 Page 9, line 14: "A record-breaking event is defined as the annual value that exceeds all the previous ones." Such definition implies a greater frequency of events at the beginning of the record. Did the authors consider identifying record-breaking events by fixing a threshold for rainfall intensity, basing on information that may be extracted by the whole record of observations (without introducing any assumption on the underlying probability distribution.*

*R#AM_C4: Page 9, line 14: it is stated that "At this stage, only nationwide record-breakings are considered, pulling up all the data together year by year." I do not understand how data were pulled together. Did the authors pool together sites with different climatic behaviours? Does this mean that record-breaking events at the local level may have been discarded?*

We are aware that to carry on analysis that involves pooling up of the data requires homogeneity controls and the wide domain under analysis does not warrant for climatic homogeneity. However, we propose a countrywide record-breaking analysis to provide a general assessment on the large rainfall amounts we are referring to when speaking about the "extremes of the extremes". Location of the record breaking also point out their distribution in space. This analysis does not allow to consider record breaking events at the local level. We will clarify this in the revised version of the manuscript. On the other hand, our research on the topic is ongoing, as we are trying to identify a meaningful threshold for selection of homogeneous extraordinary events at the country scale. This additional analysis is still under development and will require a significant amount of efforts. We have then decided to devote a different paper to this analysis, that can have implication in terms of significance of increasing trends of short duration rainstorms.

*R#1_C5 Figure 4. Excellent Figure. Page 8, lines 7-8 ("When short...appears."). I agree with your statement, nonetheless this could also be an artifact due to the color scale chosen. Is there any particular reason for your choices of min/max values for color scale of the different durations? Does the min/max values correspond to any percentile of the distribution of values, for example?*

The reviewer is right, the apparent distribution of the median values could be also an artifact of the chosen color scale (the min/max values correspond to the min/max values of the distributions). Nonetheless, the growing coefficient of variation of the national distribution of the median values when larger durations are considered seems to confirm a different distribution of the median values across the durations. These considerations are included in the revised manuscript at P8, LL12-14.

*R#AM_C3 Page 3, line 11: it is stated that "Considering that most of the provided data have been validated from the related authorities, they are considered reliable and, at first, included directly in the I-RED" It would be interesting to discuss the validation tests that have been considered by the authorities. Extreme rainfall data may be affected by relevant uncertainty, it would be useful to mention the gauging methods, what kind of checks have been considered by the authorities and so on.*
*R#4_C1 Some discussion is needed on how the rainfall data at these stations is collected, as well as how the practice of recording rainfall data over time may have changed. For example, there may be some stations that reported 24 h rainfall totals for a longer period of time than 1 h rainfall recordings. This additional discussion could be aided by a plot similar to Figure 2d for the shortest (1 h) and longest (24 h) durations. Information on the types of gauges at these stations could be critical to understanding how capable the dataset is at capturing the most extreme rainfall amounts.*

The reviewers are right. Fully detailed information on the measurement methods and on the validation of the data wold be an interesting and useful information to be associated with the data. Unfortunately, despite we can certainly states that they respect the WMO standards we can not provide this kind of information as it is not generally published by the surveying authorities. The data managed from the *SIMN* are usually from manual gauges, like the ones described in Acquaotta et al., 2016. The regional agencies then, with different timing, started to substitute the manual tipping-bucket rain gauges with automatic ones. Despite the information of the rain gauge model adopted is lacking for almost all the regions (except some rare cases, like the Veneto Region that provides a full report on the data: http://www.arpa.veneto.it/temi-ambientali/agrometeo/file-e-allegati/atlante-precipitazioni/20_Strumenti%20e%20criteri%20di%20osservazione%20e%20di%20gestione%20dei%20dati%20-

%20La%20serie%20pluviometrica%201984-2010%20dell2019ARPAV.pdf) we can suppose that the rain gauges adopted from ARPA Piemonte, described in Acquaotta et al., 2016, are representative of the ones adopted from the different regional authorities. All the data are thus recorded with a time resolution that allows to capture the extreme rainfall amounts for all the considered durations (despite sometimes for some durations some data are lacking due to different motivations, e.g., missing reading of the record from the operator, etc.). These information are reported at P2, LL20-31 of the revised manuscript.

It is also not easy to find detailed information on the validation procedures adopted, despite many of the regional authorities have undersigned the document from the Italian Insitute for Environmental Protection and Research about the validation procedure that should be adopted for the validation of meteorological databases. As the document is available only in Italian (Barbero et al, 2016) a summary is translated and reported in Appendix A and cited at P4, L3.

*R#1_C4* *Figure 2. Panels b and c shows exactly the same histogram. Please, check it.*
*R#4_C2* *Figure 2 (b) and (c): These histograms look identical. Is it true that the same number of stations that reported 24 h rainfall also reported 1 h rainfall?*

The distribution of the length classes is similar across the different duration. As it is not possible nor useful to show such similar behavior twice, figure 2 has been edited in the revised manuscript. Now panel (b) shows the number of series longer that fixed length values and, for underlying the differences across the durations, panel (c) report the number of null values per duration. Some lines on the source of the null values are added at P7, LL25-27.

*R#2_C1* *Figure 2: There is not any official abbreviation of years as "Y" so probably it would be more clear instead of t(Y) to write just "Years". Panels b and c: please decrease the size of fonts in the X-axis so it can be read more easily and also change the label to "Length (years)" or something similar as no.years is confusing (also check the panel d: do you mean years or data?). This suggestion is for any other Figure, e.g., for Figure 5 where "t(Y)" is used.*

Figure 2 and 5 of the revised manuscript have been amended according to the reviewer's suggestions.

*R#2_C2* *Analysis presented in Figure 4. There is a dense network, more than sufficient to provide kriging estimates for the whole Italy. It would much more useful in my opinion not only because dots may overlap but because you will provide estimates also in places where there is no information. So, I would suggest to construct kriging maps of the statistics analyzed.* *E* *Some supplementary material on the spatial variability of rainfall in Italy (maps) will also be more than welcome.*

Considering the exploratory nature of our analysis we did prefer to provide point statistics, which –at this stage- indicate the main features of rain variability at the country scale but do not address the goal of providing robust estimates in ungauged areas. Considering the many problems we have faced in the regional analysis in the North-West of Italy (Libertino et al, 2018) the uneven distribution of the rain gauges in space and the high variability of record length make the whole kriging analysis a challenge deserving a separate work. In partially following the reviewer's suggestion, however, interpolated maps will be added as supplementary material to more clearly represent rainfall indices variability over space.

*R#2_C3* *Mean is quite robust in general, yet here you prefer only the median. Of course, it can be affected by outliers yet common methods of fitting distributions, e.g., product moments or L-moments are using mean values. So, in my opinion you should provide also the maps of the mean value.*
*R#2_C4* *If I understood well you have estimated the mean values of L-CV, L-skew and L-kurt of all duration. Of course you are dealing with maxima and we are expecting the shape characteristics to be close yet this is not necessarily true. If indeed these summary statistics are close among the different duration, please report it or else provide different maps for each duration.*

The reviewer is right: most of the common methods of fitting distributions consider the mean as best descriptor of the central tendency. However, in this descriptive phase we wanted to represent the variability of the central tendency of the series as unbiased as much as possible with respect to the outliers. On the other hand, we admit information of the mean can be of interest for the readers: therefore, we have added a map of the mean of the extremes it in the supplementary material as reported at P8, L11 of the revised manuscript. Brief considerations on the behaviour of the L-moment statistics across the different durations have been added in the revised version

of the manuscript at lines P8, LL18-19 and P9, L1, and the maps of the ratios for the different durations added in the supplementary materials.

**R#2_C5**  *It is not clear to me if the maxima values have emerged from a sliding-window process or from a fixed-block (non-overlapping).  In the latter case the user of this database should know this fact in order to correct the data by the Hershfield factor. Please comment on that and clarify.*

Annual maxima have been collected with a sliding-windows procedure. The information is added in the revised version of the manuscript at P2, L20.

**R#2_C6**  *You can use plain text for L-CV. In Figure 4 is plain while in the text you are using Italics.  Please re-check the text for minor typos, e.g., line 26, p6 replace "an unique" with "a unique".*

**R#4_C5**  *There are a few typos in the paper, including the following:  Page 1 Line 24:  "dismantledment" should be "dismantlement", Page 1 Line 25: "stucked" should be "stuck", Page 2 Line 2: "metodologies" should be "methodologies".*

The typos reported by the reviewer have been amended and the manuscript double-checked for more typos.

[revised manuscript text omitted]

---

## Author Response (AR2)

Responses to Editor's Comments
HESS-2017-752
**Technical note: Space-time analysis of rainfall extremes in Italy: clues from a reconciled dataset**
Libertino, Andrea*; Ganora, Daniele; Claps, Pierluigi
*andrea.libertino@polito.it

The comments from the editor have been reproduced below, interspersed with our indented responses

Page 2, line 17: (PRIN 2005) does not appear in the list of references

> PRIN is an acronym for "Italian Research Projects of National Relevance", a projects call funded by the Italian Ministry of Education and Research. As it was not clear in the previous version of the manuscript, it is explained in the revised version at P2, L15.

Please, capitalize the word "Figure" when used in the text, e.g. Page 3, line 2 & Page 5, line 12 & Page 7, lines 8, 21, 25, 28, and 31 & Page 8, lines 8, 18 & Page 10, line 4.

Page 9, line 1 & 4 - it would be better to use "Figure 4f" instead of "Panel (f)", and "Figure 4g and 4h" instead of "panels (g) and (h)".

Page 10, line 17 - use "Figure 5c" rather than "Panel (c)".

Page 11, lines 17 & 22: capitalize "Table 1".

> All the comments of the editor have been amended in the revised version of the manuscript.

[revised manuscript text omitted]